# Entomological surveillance on *Aedes aegypti* during covid 19 period in Cape Coast, Ghana: Risk of arboviral outbreaks, multiple insecticide resistance and distribution of F1534C, V410L and V1016I *kdr* mutations

Joana Ayettey[1], Aikins Ablorde[2], Godwin K. Amlalo[3], Ben A. Mensah[1], Andreas A. Kudom[1] *

1 Department of Conservation Biology and Entomology, University of Cape Coast, Cape Coast–Ghana, 2 Center for International Health, Ludwig Maximilian University of Munich, Germany, 3 Department of Parasitology, Noguchi Memorial Institute for Medical Research, University of Ghana, Accra Ghana

* akudom@ucc.edu.gh

## Abstract

### Background

The study assessed the risk of transmission of *Aedes*-borne arboviruses in a community at Cape Coast during the Covid-19 restriction period in 2020 based on entomological indices. The spatial distribution of insecticide resistance was also assessed in *Ae. aegypti* population from Cape Coast.

### Methods

Three larval indices were calculated from a household larval survey in 100 randomly selected houses. WHO susceptibility bioassay was performed on female adult *Ae. aegypti* that were reared from the larvae collected from household containers and other receptacles located outside houses against four insecticides. The mosquitoes were also screened for F1534C, V1016I, and V410L *kdr* mutations.

### Results

The estimated larval indices in the study community were House index– 34%, Container index– 22.35%, and Breteau index– 2.02. The mosquito population was resistant to Delta-methrin (0.05%), DDT (4%), Fenitrothion (1%), and Bendiocarb (0.1%). A triple *kdr* mutation, F1534C, V410L and V1016I were detected in the mosquito population.

### Conclusion

The study found the risk of an outbreak of *Aedes*-borne diseases lower in the covid-19 lock-down period than before the pandemic period. The low risk was related to frequent clean-up exercises in the community during the Covid-19 restriction period. Multiple insecticide

**Data Availability Statement:** The authors confirm that all data underlying the findings are fully

available without restriction. All relevant data are within the paper and its Supporting Information files.

**Funding:** The author(s) received no specific funding for this work.

**Competing interests:** The authors have declared that no competing interests exist

resistance couple with three *kdr* mutations detected in the study population could affect the effectiveness of control measures, especially in emergency situations. The study supports sanitation improvement as a tool to control *Ae. aegypti* and could complement insecticide-based tools in controlling this vector.

## Author summary

The study assessed the risk of transmission of *Aedes*-borne arboviruses in a community in Cape Coast during the Covid-19 restriction period in 2020. Three larval indices were calculated from the household larval survey in 100 randomly selected houses in the community. Insecticide susceptibility of the female adult *Ae. aegypti* to four insecticides was performed. The results showed that the risk of a potential outbreak of *Aedes*-borne diseases in the Covid 19 restriction period was much lower than in 2017 when a similar assessment was done. The low risk was related to frequent clean-up exercises in the community during the Covid-19 restriction period. The mosquito population was resistant to deltamthrin, DDT, bendiocarb and fenitrothion insecticides. However, resistance level was higher in the mosquitoes collected in houses than the mosquitoes collected outside houses. Insecticide use in houses may have contributed to the different level of resistance observed. The study detected F1534C, V1016I and V410L *kdr* mutations in *Ae. aegypti* population. The combination of these mutations is known to generate high levels of pyrethroid resistance in the mosquito. The study agrees with sanitation improvement as a tool to control *Ae. aegypti* and could complement insecticide-based tools in controlling this vector.

## Introduction

*Aedes aegypti* is a medically important mosquito due to its involvement in the transmission of several arboviruses to humans. In the last decade, there have been several epidemics of *Ae. aegypti*-borne diseases in many countries. In 2016, two linked urban yellow fever outbreaks were reported in Luanda (Angola) and Kinshasa (the Democratic Republic of the Congo) in Central Africa [1]. In the same year, large dengue outbreaks in the Region of the Americas reported more than 2.38 million cases [1]. In 2015–2016, the Zika virus disease outbreak was also reported in some countries in South America. Sadly, as the world battles with the Covid-19 pandemic, there is also a surge of *Aedes*-borne arboviral diseases in some countries. For instance, between January and June 2020, about 1.6 million dengue cases, 37,000 chikungunya cases, and 7000 Zika cases have been reported in the region of the Americas [2]. The combined impact of both Covid-19 and outbreaks of *Aedes*-borne arboviral diseases could have potentially devastating consequences, particularly in low-resource countries. This calls for advanced preparation by endemic countries against any *Aedes*-borne disease outbreak.

In urban areas, *Ae. aegypti* normally breeds in water storage containers as well as other man-made receptacles that can hold water [3,4]. There is a serious concern that the use of these storage receptacles could increase in this pandemic period. Because it is anticipated that the demand for water in households would increase due to some Covid-19 mitigation measures such as regular washing of hands and other hygienic practices that require a constant flow of water. In fact, for this reason, the Ghana government declared a 'free water for all' initiative on 5 April 2020 where water was supplied for free in the country as a relief measure

during the Covid-19 restriction period [5]. An increase in water demand, particularly in households with limited access to pipe water, may lead to increased use of water storage containers. The danger is that if the containers are not managed well, they could serve as a breeding place for *Ae. aegypti* and other container-breeding mosquitoes. A similar situation was reported during a drought period in Australia in 2008 [6]. Household water storage practices in response to drought resulted in new habitats for domestic container-breeding mosquitoes [6]. Such conditions could affect the epidemiology of mosquito-borne diseases and their control.

In Ghana, about 49% of households use pipe-borne water as the main source of water for general use. However, only 10.6% of households have piped-borne water inside their premises (GLSS7). Before the Covid-19 outbreak, more than 80% of households in Ghana stored water in plastic containers/buckets (GLSS7). The high density of *Ae. aegypti* population in Ghana has been attributed to water storage practices and poor sanitation conditions [4,7,8]. With the anticipated increase in water storage practices in the Covid-19 period, the question is, would the risk of a potential outbreak of *Aedes*-borne arboviruses increase as compared to the period before the pandemic?

Emergency control measures presently depend on insecticide-based strategies, especially, pyrethroid insecticides. Unfortunately, the emergence and development of insecticide resistance among mosquito vectors could affect the effectiveness of insecticide-based strategies. For example, evidence of reduced effectiveness of space spray for control against pyrethroid-resistant *Ae. aegypti* has been reported in the island of Martinique [9]. This calls for continuous monitoring of insecticide resistance status of vector populations and development of resistance management strategies. Management of insecticide resistance requires understanding of the major resistance mechanisms in the vector population as well as patterns of cross resistance among various classes of insecticides. Two major physiological mechanisms involved in resistance among vector mosquitoes are 1) alteration of target sites, and 2) increased metabolic detoxification. However, knockdown resistance (*kdr*) mutation in the voltage-gated sodium channel, which is responsible for conferring cross resistance between pyrethroids, and DDT are the most studied mechanism in *Ae. aegypti* [10–12]. About ten major *kdr* mutations have been detected in *Ae. aegypti* [13]. The widely distributed *kdr* mutation in *Ae. aegypti* population in Africa is F1534C mutation, which together with V1016I were first detected in Africa from *Ae. aegypti* population from Ghana [14]. The mutations have since been found in different African countries [15–17]. As part of the entomological surveillance on *Aedes aegypti* during covid 19 period in Cape Coast, we investigated the spatial distribution of insecticide resistance in the study area. The question is, in case of an outbreak, is the current insecticide-based mosquito control strategy completely reliable for emergency control? Answers to these questions may help prepare in advance against any potential disease outbreaks from *Ae. aegypti* in the country.

## Materials and methods

### Study area

The study was conducted in Cape Coast (5˚06′N 1˚15′W; 5.1˚N 1.25˚W), a coastal city of approximately 122 km$^2$ of land, which is about 165 km west of Accra, the capital of Ghana. Cape Coast has a population of about 189,925 to the 2021 Census (Ghana Statistical Service). The major rainy season is between May and July with mean monthly relative humidity ranging from 85 to 99%. The total monthly rainfall and mean daily maximum temperature for 2017 and 2022 have been summarized in Fig 1. The city has several coastal thickets (remnant forests) containing several species of primates [18].

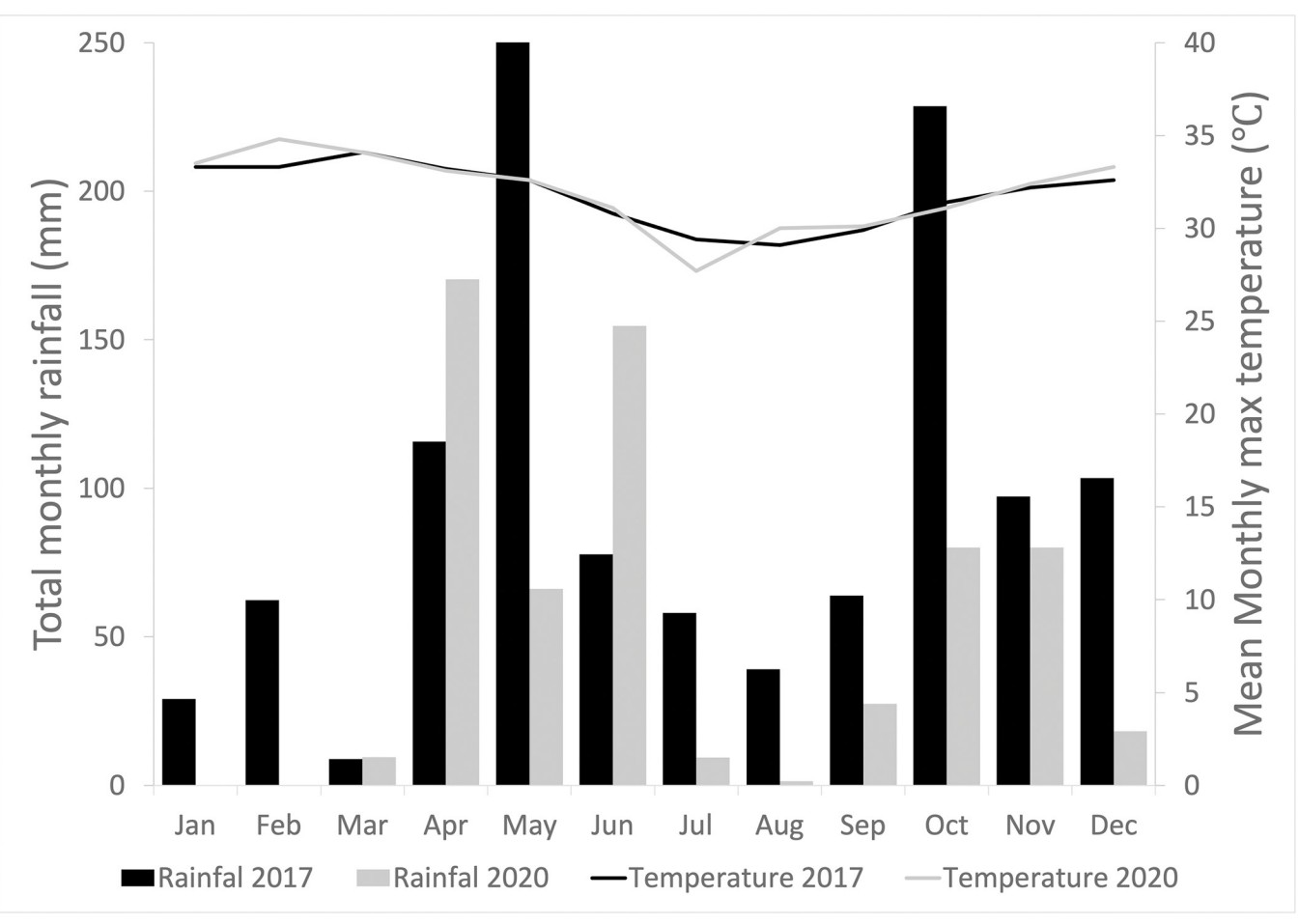

**Fig 1. Total monthly rainfall and mean monthly maximum daily temperature recorded for Cape Coast in 2017 and 2020.**

### Covid 19 situation in Ghana

In Ghana, there have been 168,580 confirmed cases of Covid-19 with 1,459 deaths as of August 2022 [19]. The first case of Covid 19 was reported in March 2020 and the country saw a sharp rise in infections in many parts of the country. Initially, the government of Ghana banned all public gatherings, including conferences, workshops, funerals, festivals, political rallies and church activities. Basic and high schools, as well as universities, were closed to reduce the spread of the virus. The government also imposed 21-day partial lockdown restrictions on Accra, Kumasi, and Kasoa, after they emerged as hotspots for the spread of the disease. This period is what this study described as Covid 19 restriction period. This study was conducted from July to September 2020 in Duakor, a community in Cape Coast. During this period, schools were still closed. However, the final year students of both the universities and the high schools were back to school to write their final examinations. In that period, piped water was supplied for free in the country by the government of Ghana as a relief measure against the impact of Covid 19. The risk of transmission from the present study was compared with a similar study conducted in 2017 in the same area. The total rainfall during the present study period, July to September 2020 and the previous study April to July 2017 were 12.8 mm and 125.6 mm respectively.

## Household larval survey

Duakor is a suburb of Cape Coast with a poor structural layout. Most households depend on rainwater, communal taps, and wells. Containers are of great importance in this area for the storage of water. Following the procedures outlined by WHO [20], a survey frame of houses in Duakor was obtained and a simple random sampling method was employed to choose 100 houses from the community for the study. Pre-adult mosquito stages were collected from the selected households during the entomological survey. The number of receptacles in and around each house was recorded and the water containers were searched for immature mosquito forms, which were collected for identification in the laboratory. The water containers found were recorded as 1) those used for water storage (category A) and 2) discarded containers (category B). From the house data, House Index (HI), Container Index (CI), and Breteau Index (BI) were calculated [20]. The House Index (HI) was expressed as the percentage of houses infested with *Aedes* larvae and Container index (CI) as the percentage of containers infested with the larvae while Breteau index (BI) was expressed as the number of positive containers per 100 inspected houses.

## Insecticide resistance bioassay

The level of insecticide resistance of *Ae. aegypti* to Deltamethrin (0.05%), DDT (4%), Fenitrothion (0.1%) and Bendiocarb (0.1%) were determined following WHO standard protocols using the WHO impregnated papers and test kits [21]. The first emergence (F0) of *Ae. aegypti* collected as larvae from household water storage containers (HWSC) located inside houses as well as abandoned car tires (ACT) and abandoned plastic containers found outside the residential houses were used for the susceptibility bioassay. About twenty-five, unfed 3-to-5-day-old females were exposed to each insecticide with four replications for each of the three vector populations. The laboratory conditions for the bioassay were 29 ± 1˚C and relative humidity of 75 ± 10%. After the exposure, mosquitoes were transferred to holding tubes, fed with 10% sugar solution and knockdown read after one hour. Total death or mortality was read after 24 hours. For control, mosquitoes were exposed to paper without any insecticides. The mortalities caused by the insecticides were expressed in percentages.

## Screening for *kdr* mutations

DNA of hundred adult female mosquitoes from the three populations (HWSC, ACT, APC) was extracted using the QuantaBio Extracta kit according to the manufacturer's direction. Knock-down resistance (*kdr*) mutations were screened using qPCR melting curve analysis. The Saavedra-Rodriguez et al. [22] (V410L) and Estep et al. [23] (V1016 and F1534C) protocols were optimized and modified for this work. For V410L detection, a total of 20μL reaction contained 0.05μM of V410fw and L410fw primers and 0.1μM of primer 410rev, 9.5μL of 2x Sybr Hi-Rox Mix (Bioline), 1μL of genomic DNA and DNase-free water. V1016I detection was done in a 20μL reaction consisting of 8.2μL of 2x Sybr Hi-Rox Mix (Bioline), 0.15μM of Val1016f primer, 0.2μM each of lle1016f and lle1016r primers, 2μL DNA template and DNase-free water. The cycling condition for this procedure is 95˚C for 3minutes, 40x (95˚C: 10sec, 60˚C:10sec,72˚C: 30sec) 95˚C with the melting condition of 65˚-95˚ inc 0.2˚C per 10sec. Each 20μL reaction for V1016I consisted of 8μL of 2x Sybr Hi-Rox Mix (Bioline), 0.15μM of Val1016f primer, 0.2μM each of lle1016f and lle1016r primers, 2μL DNA template and DNase-free water, with the same cycling conditions as the V410L. F1534C was screened in a reaction volume of 20μL containing 8μL of 2x Sybr Hi-Rox Mix (Bioline), 0.3μM of Cys1534+ primer, 0.3μM each of Phe1534+ and 1534- primers, 2μL DNA and DNase-free water. The cycling conditions for F1534C included 3min at 95˚C, 37 cycles of (10 sec at 95˚C, 10 sec at 57˚C, 30

sec at 72˚C) and 95˚C for 10 seconds. A melting curve analysis was performed at the end of the PCR cycle at 65˚C—95˚C. The mutant gene (resistant) and wild type (susceptible) respectively produced melting curve peaks at 83˚C and 86˚C for V410L, 80˚C and 86˚C for V1016I, and 86˚C and 82˚C for F1534C. A chi-square test for deviation from Hardy-Weinberg equilibrium of the genotype frequency distribution was performed using Gene-Cal.

## Results

### Household container survey

In the houses surveyed, 850 water-holding containers were found in 100 houses. About 22% of the containers in 32 houses were positive for mosquito larvae (Fig 2). A total of 2829 larvae and 481 pupae were collected from the positive containers (Table 1). Water storage containers in the houses (Category A containers) harboured 84% and 96% of total larvae and pupae that were collected respectively. Pupal productivity was highest for buckets (57.4%) whereas pupal productivity for cement tanks and barrel/drums were 29.9% and 12.8% respectively (Fig 3). *Aedes* larvae constituted 66.8% of the total larvae with 27.4% being *Culex* while 4.3% and 1.45% were *Lutzia* and *Anopheles* larvae respectively. All the 481 pupae collected were *Aedes* mosquitoes. The estimated larval indices in the study community were House index– 34%, Container index– 22.35%, and Breteau index– 2.02.

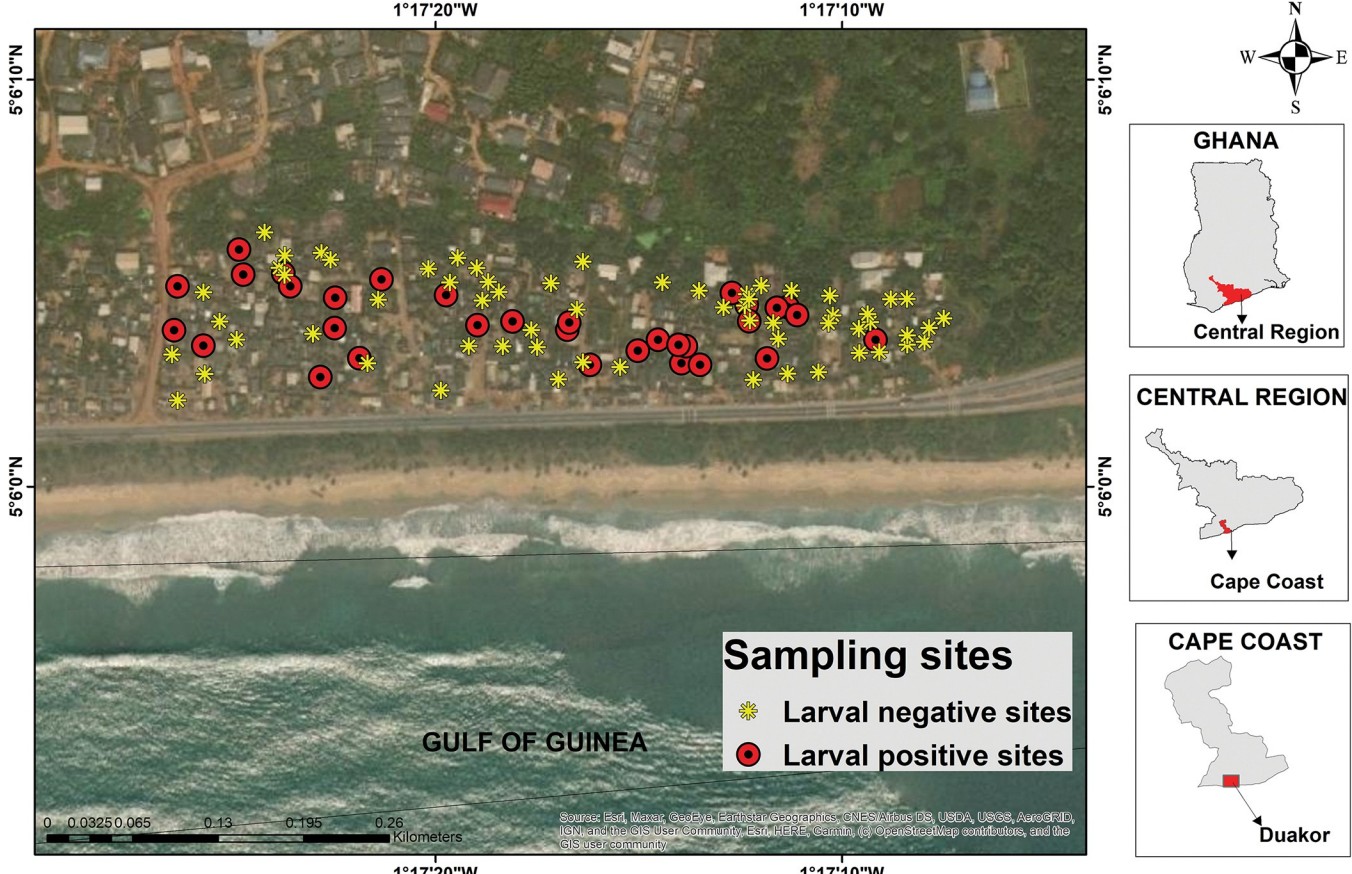

**Fig 2. Spatial distribution of households that were positive or negative for mosquito larval-infested items.** The shapefiles were created with ArcGIS from GPS coordinates collected in the field, and the base map was sourced from OpenStreetMap (https://www.openstreetmap.org).

**Table 1. Number of containers found in households with infestations of mosquito larvae and the total number recorded.**

| Items Category | Specific Items | Number of Items (% infested) | Total larvae collected | Percentage of larvae: | | | | Total pupae collected |
|---|---|---|---|---|---|---|---|---|
| | | | | *Aedes* | *Culex* | *Lutzia* | *Anopheles* | |
| Category A: household water storage containers | bucket | 306 (2.9%) | 65 | 100 | - | - | - | 265 |
| | 5–20 L plastic containers | 18 (61%) | 569 | 37.4 | 59.8 | 2.8 | - | 0 |
| | Concrete tank | 39 (51%) | 1088 | 57.7 | 33.6 | 4.9 | 3.8 | 138 |
| | barrel/drum | 54 (19%) | 656 | 95.9 | 0 | 4.1 | 0 | 59 |
| | bowls/jars | 24 (0%) | 0 | - | - | - | - | 0 |
| Total | | 441 (11%) | 2378 | 64.6 | 29.7 | 4.0 | 1.7 | 462 |
| Category B: discarded items | tires | 13 (39%) | 241 | 88.8 | - | 11.2 | 0 | 15 |
| | coconut shell | 313 (0%) | - | - | - | - | - | - |
| | bowls/jars | 15 (33%) | 100 | 30.0 | 70.0 | - | - | 0 |
| | bottles | 170 (76%) | 110 | 100 | - | - | - | 4 |
| Total | | 511 (27%) | 451 | 78.5 | 15.5 | 6.0 | 0 | 19 |

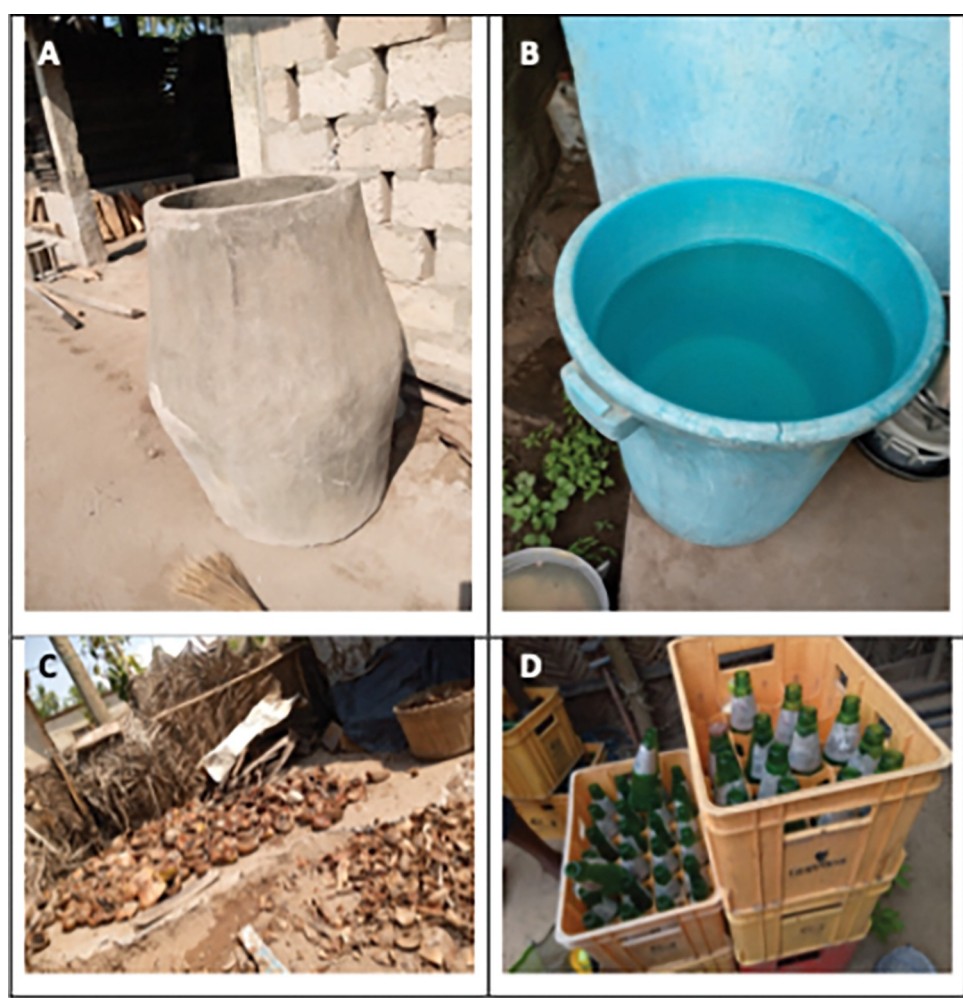

**Fig 3.** Different containers infested with *Aedes aegypti* larvae in Duakor community. Cement tank (A), drum (b) of categeory A items and coconut shells (C) and bottles (D) of category B items.

### Insecticide resistance in the *Aedes aegypti* population collected from household water storage containers (inside houses) and abandoned automobile tires and plastic containers (outside residential houses)

In total, *Ae. aegypti* from the study community were resistant to the four insecticides (Fig 4). However, resistance differed in the *Ae. aegypti* collected from the different breeding habitats (Fig 5). Besides deltamethrin, *Ae. aegypti* collected from the household water containers (HWSC) were more resistant to DDT (P = .002), Fenitrothion 1% (P = .005), and Bendiocarb 0.1% (P = 0.002) than those collected from the abandoned automobile tires (ACT) or plastic containers (APT) outside residential houses. It was more pronounced in Bendiocarb and DDT insecticides (Fig 5). A mean %mortality of 32 ± 20 and 24 ± 14.9 of Bendiocarb and DDT were recorded for the vector population collected from household water storage containers, which were significantly lower than 94 ± 2.8 and 74.7 ± 9.2 for the population collected from automobile tires and 88 ± 45.7 and 60 ± 12.7 for those collected from abandoned plastic containers against the same insecticides respectively. Similarly, the vector population collected from outside the residential houses were almost susceptible (ACT: 97.3 ± 4.6; APC: 98 ± 2.3) to Fenitrothion whereas the insecticide caused a mean %mortality of 74 ± 12.4 to the vector population collected from HWSC.

Among the 100 mosquitoes, 98, 97 and 96 were successfully genotyped for the F1534C, V410L and V1016I *kdr* mutations, respectively (Tables 2 and S1). The homozygote mutant at codons 1016 and 410 was absent whereas the wild type of F1534C was rarely (2%) observed. F1534C mutation was widely distributed in the study population with the heterozygote F / C1534 genotype mutant dominating in the population. Higher allelic frequency of F1534C mutation was observed in the *Ae aegypti* population collected from HWSC compared to those collected outside residential houses (ACT and APC) (Table 2) ($X^2$ = 33.93, df = 2, p < .0001).

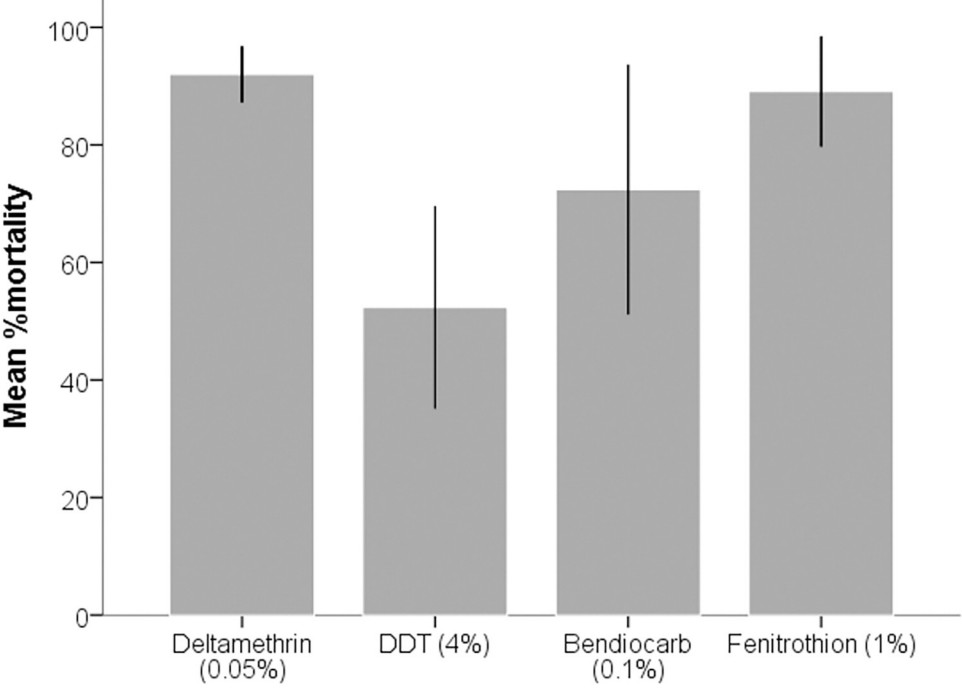

**Fig 4. Susceptibility of female adult *Aedes aegypti* from Cape Coast to Deltamethrin (0.05%), DDT (4%), Fenitrothion (0.1%) and Bendiocarb (0.1%) using WHO test kits.**

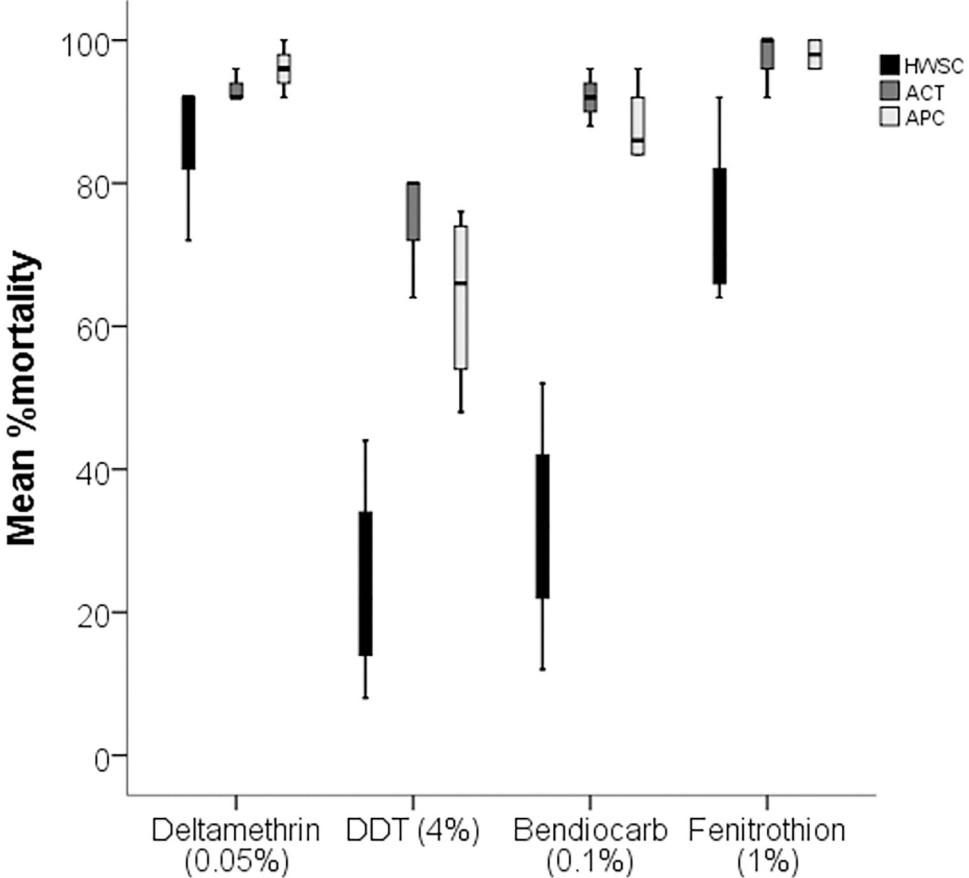

**Fig 5. Susceptibility of female adult *Aedes aegypti* collected from household water storage containers (HWSC) (inside houses) and those collected from abandoned car tires (ACT) and abandoned plastic containers (APC) (outside residential houses) to Deltamethrin (0.05%), DDT (4%), Fenitrothion (0.1%) and Bendiocarb (0.1%) using WHO test kits.**

However, V1016I *kdr* mutation was absent in the population from the HWSC whereas low allele frequency was observed in the populations collected outside the residential houses (APC and ACT). Nine genotypes across the three *kdr* mutations were identified from the 93 *Ae. aegypti* population from Cape Coast (Fig 6). The genotype FC/VL/VV was the most frequently observed (37%, n = 93). Individuals that were homozygous for the three *kdr* mutations were absent. Overall, the genotype frequency distribution of V1016I was consistent with Hardy-Weinberg equilibrium ($X^2 = 0.233$, df = 1, p = .63). However, the genotype frequency distribution of F1534C and V410L were not in Hardy-Weinberg equilibrium ($X^2 = 7.6$, df = 1, p = .006; $X^2 = 28.3$, df = 1, p < .0001).

**Table 2. V1016I, V410L and F1534C genotype numbers and the allelic frequency of the I, L and C mutations of *Ae. aegypti* collected from different breeding habitats.**

| Source of mosquito | V1016I | | | V410L | | | F1534C | | | Kdr allele frequencies | | |
|---|---|---|---|---|---|---|---|---|---|---|---|---|
| | V/V | V/I | I/I | V/V | V/L | L/L | F/F | F/C | C/C | I | L | C |
| Household water storage containers | 23 | 0 | 0 | 10 | 12 | 0 | 0 | 2 | 22 | 0 | 0.27 | 0.96 |
| Abandoned car tires outside houses | 43 | 7 | 0 | 9 | 41 | 0 | 2 | 32 | 16 | 0.07 | 0.41 | 0.64 |
| Plastic containers outside houses | 21 | 2 | 0 | 10 | 15 | 0 | 0 | 16 | 8 | 0.04 | 0.30 | 0.67 |
| Total | 87 | 9 | 0 | 29 | 68 | 0 | 2 | 50 | 46 | 0.05 | 0.35 | 0.72 |

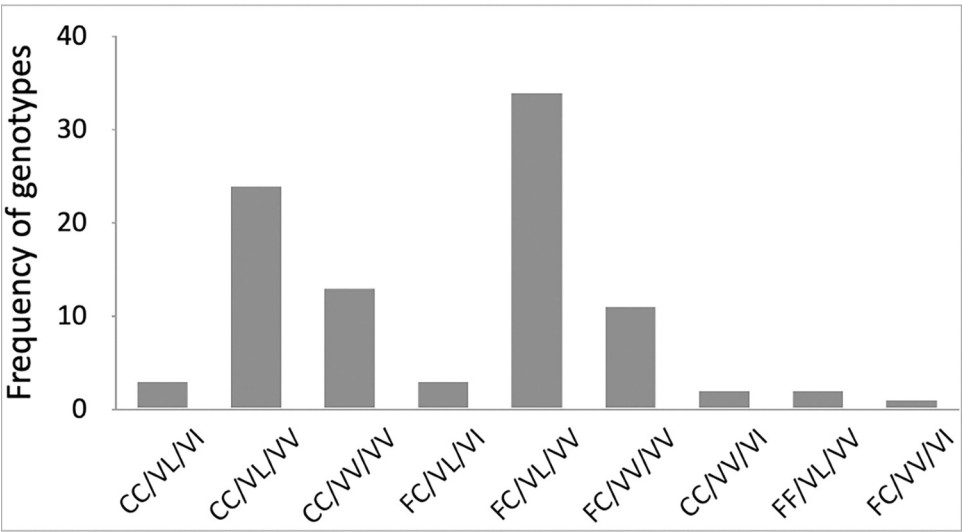

**Fig 6. Frequency of the combined V1016I, V410L and F1534C genotypes of *Aedes aegypti* from Cape Coast.**

## Discussion

The study assessed the risk of transmission of *Aedes*-borne arboviruses in the Duakor community in Cape Coast during the Covid-19 lockdown period in 2020 based on three larval indices: container index, house index and Breteau index. The results were compared with a previous study conducted in 2017 in the community before the Covid-19 pandemic. The assumption was that Covid-19 mitigation strategies could increase the demand for water, which in turn, might influence significant changes in householder behavior towards water storage practices. A change in behavior that would lead to increased use of water storage containers may promote the breeding of *Ae. aegypti* and other container-breeding mosquitoes. This may lead to an outbreak of mosquito-borne diseases. Among the three indices determined in the study, only the container index marginally exceeded the WHO threshold. Both Breteau and house indices were below the WHO threshold. Contrary to the assumption made, the result on the larval indices suggested a lower risk of transmission of *Aedes*-borne diseases in the Covid-19 restriction period compared to the 2017 survey period.

In the previous study in Cape Coast in 2017, discarded and abandoned containers/receptacles (category B) were important breeding grounds in all the four communities and accounted for 60% of the infested containers. Water storage containers were less important in the three communities that had better access to pipe water. However, water storage containers together with discarded items (category B items) were equally important in Duakor community (the study area of the present study), which had the least access to piped-borne water. Duakor accounted for 77% of infested household water storage containers among the four study communities [4]. In this study, fewer category B items were observed to have been infested with mosquito larvae, and this may have contributed to the lower larval indices compared to the study before the pandemic [4]. The category B containers contributed only 4% to the total pupae collected in the study. During the Covid-19 restriction period, there were frequent clean-up exercises in both public and residential areas [24] leading to the removal of discarded items from the compounds. This may have improved the sanitation condition in the study community resulting in fewer mosquito-infested category B items. The total precipitation during the present survey period (July-September) was extremely lower than the survey period (April-July) in the previous study (Fig 1). This could also have contributed to the differences in

the infested category B items recorded between the two studies. However, we think precipitation might not have contributed much on the infestation of category B items in the present study. For precipitation to influence the level of infestation of category B items, discarded containers must first be available. Unlike the previous study, discarded items were extremely low in this study due to the clean-up exercise and other factors mentioned earlier. It is worth mentioning that the bottles and coconut shells, which formed 95% of category B items in the present study were technically not discarded or abandoned items (Fig 3C and 3D). The bottles and the coconut shells were packed because of restricted movement during that period. They were waiting to be transported to their destinations.

Furthermore, the Government of Ghana's free water initiative greatly improved water accessibility in Ghana during the Covid-19 restriction period [25] and this initiative may have minimized water stress in the community. The limitation of this study was that the larval survey was conducted only in one community. A larval survey from different communities with different levels of accessibility to pipe-borne water may have provided a better picture of the situation than one community. However, Covid-19 restrictions made it very difficult to include other communities during the study period. Another limitation of the study was that the infectious status of the mosquitoes collected were not determined. Although, entomological surveillance tools such as the use of larval or pupal indices to assess the risk of outbreak of *Aedes* borne diseases remain important, especially in low endemic areas like Ghana, information on the infectious status of the mosquitoes could have given a better picture of the situation. A future study that would determine the infectious status of the vector together with the entomological indices may give a better risk assessment of the situation in study area.

Notwithstanding, health education on mosquito prevention in the community is still needed. All the houses that were surveyed in the study community had water storage containers that were either partially covered or not covered at all. The practice of storing water has been reported to be a reason for the increase in mosquito productivity [26,27]. It was therefore not surprising that the majority (86%) of the mosquitoes collected in the study came from water storage containers. Health education such as the promotion of the use of lids to cover long-term water storage containers can drastically reduce the presence of *Aedes* larvae in the community [28]. The promotion of this simple technique could have been incorporated into the Covid-19 public health campaigns. A recent study in Cape Coast found that intensive public health education during the pandemic has greatly improved health knowledge among the population [29]. However, the improved knowledge was related to chronic diseases, nutrition, hygiene, and risky health behaviors. Improving the knowledge of anti-mosquito strategies in the community may also help further lower the population density of *Aedes* mosquitoes and ultimately the risk of an outbreak of *Aedes*-borne diseases.

The most productive containers are identified by determining the relative contribution that a particular container type makes to the overall production of *Aedes* pupae [20]. The most productive containers from this study were concrete tanks and medium to large plastic containers. However, buckets were the most productive contributing to more than 50% of the pupae collected in this study. This is consistent with the findings from other countries where concrete washbasins, drums and buckets are the most productive household containers for *Aedes* mosquitoes [30–32]. Mosquito larvae feed primarily on aquatic microorganisms that colonize detritus in breeding habitats and the chemicals produced by the microorganisms also influence the oviposition behaviors of adult female mosquitoes [33]. Under normal conditions, it may take some time for microorganisms to colonize and build up their population in water-filled containers. It is therefore not surprising that this study and many other findings have shown containers (e.g., cement tanks, barrels) that are used to store water for a long period to be more productive than containers (e.g., bowls, jars) that are mostly used to store water for short

period [27,30,32]. Targeting these containers for vector control may greatly reduce the population of *Aedes* in the community.

Although *Ae. aegypti* constituted the major mosquito larvae collected, *Culex*, *Anopheles* and *Lutzia* mosquitoes were also found in the household containers. These species are known to co-exist in breeding habitats in Ghana. *Lutzia* is a well-known mosquito predator whilst *Aedes*, *Culex*, *Anopheles* are also known to exhibit interspecies predatory activities [34,35]. Indeed, the fourth instar *Lutzia* larva can consume up to 24 fourth instar *Aedes* larvae per day [36]. In a mixture of different mosquito larvae, Appawu et al [34] reported that *Lutzia* larvae exhibited a significant preference for *Ae. aegypti* larvae compared to *An. gambiae* s. I. and *Cx quinquefasciatus*. Interaction of these mosquito larvae in the containers may have a significant influence on the resulting adult populations of *Ae. aegypti*. Further study is needed to elucidate the impact of such complex interaction among mosquito larvae in household containers on *Aedes'* productivity.

Chemical control remains an important part of most control measures against mosquito vectors. In this study, *Ae. aegypti* population from Cape Coast was highly resistant to deltamethrin (pyrethroid), DDT (organochlorine) and bendiocarb (carbamate). But exhibited moderate resistance to fenithrothion (organophosphate). Resistance of the mosquitoes to the different classes of insecticides is worrying and could affect the efficacy of insecticide based control tools [9]. The level of resistance to deltamethrin and DDT is consistent with the results from other parts of the country [4,14,37]. However, the high level of resistance to carbamate (bendiocarb) was unexpected. Unlike, pyrethroid insecticides, carbamate insecticides are not normally employed in household vector control tools. Furthermore, urban agricultural with its associated insecticide use are not important activitiy in Cape Coast Metropolis. Thus, the source of resistance particularly to carbamates is not clearly known. In fact, the source of resistance in *Ae. aegypti* population from Ghana and many African countries remains less obvious [12]. Nonetheless, domestic use of insecticides could be a very important source, particularly for pyrethroid insecticides [38–40]. The result from the insecticide bioassay showed that the mosquitoes collected from containers located inside houses were more resistant to the insecticides than the mosquitoes collected from containers located outside houses (Fig 4). This differential resistance could be explained by the heavy use of insecticides in houses [38–40]. Most houses depend on daily use of insecticide-based tools such as insecticide treated nets and mosquito coils for protection against mosquito bites. This could put *Ae. aegypti* that lives and oviposit in containers located in houses under higher insecticide selection pressure than the mosquitoes that breed in containers outside houses. Understanding the spatial distribution of insecticide resistance among the mosquito population may help improve resistance management strategies.

In this study, F1534C, V1016I, and V410L *kdr* mutations were detected from the *Ae. aegypti* population from Cape Coast. Single or multiple of these *kdr* mutations have been associated with resistance to different pyrethroid insecticides and DDT [38,41,42]. F1534C mutation was previously detected in Cape Coast and other parts of Ghana and remains the most widespread mutation in *Ae. aegypti* population in Africa [43]. We report for the first time the detection of V410L and V1016I mutations in *Ae. aegypti* population outside Accra, Ghana. In the first report of the detection of V1016I mutation in Africa about six years ago [14], a single *Ae. aegypti* mosquito from Accra (Ghana) was found with the mutation. Thus, it is alarming to observe the spread of the mutation together with the V410L in Cape Coast within this short time. The combination of F1534C and V1016I have been shown to generate high levels of resistance to pyrethroid [22,41]. In fact, Vera-Maloof et al. [41] suggested that high pyrethroid resistance in *Ae. aegypti* requires the sequential evolution of F1534C and V1016I mutations. It is for this reason that the high frequency of the F1534C mutation recorded in this study in

addition to the V1016I mutation is worrying. However, the contribution of V410L to the pyrethroid resistance remains unclear. This mutation was recently detected in a population from Accra [44] as well as Ghana's neighboring countries of Cote d'Ivoire [15] and Burkina Faso [38]. However, all the three studies did not find the V410L contribution to pyrethroid resistance in their respective countries. Nevertheless, V410L alone or in combination with the F1534C mutation has been shown to reduce the sensitivity of mosquito sodium channels expressed in *Xenopus* oocytes to pyrethroids [45]. A recent study found high frequencies of the V410L *kdr* mutation to be associated with pyrethroid resistance and its combination with F1534C and V1016I was also found to influence the survival of *Ae. aegypti* after exposure to pyrethroid insecticide in a field cage tests [46].

Similar to the results from the bioassay, higher frequency of the resistant allele of F1534C mutation was found in the mosquito population collected inside houses than those collected outside the houses (Table 2). This supports the earlier suggestion that household use of insecticide may be contributing to the selection of resistance in the vector population. However, the three *kdr* mutations detected in this study cannot fully explain the multiple insecticide resistance found in the mosquito populations from Cape Coast. The resistance to bendiocarb and fenitrothion in the vector population indicates the existence of other important mechanisms. Previous study in Cape Coast detected metabolic resistance through elevated activity of mixed-function oxidase, esterase and glutathione-*S*-transferase from biochemical assays in *Ae. aegypti* population [4]. Similar result has also been reported in *Ae. aegypti* population from Accra [44]. This mechanism could explain some of the resistant phenotype found in this study. Further study is needed to elucidate all the important resistance mechanisms and the potential source of resistance for *Ae. aegypti* in the study area.

## Conclusion

The study found the risk of an outbreak of *Aedes*-borne diseases lower in the covid-19 lockdown period than before the pandemic period. Although the study sample is very limited, valuable lessons could be drawn from it concerning the control of *Ae. aegypti*. In the previous study in the community, a high number of discarded items were infested with mosquito larvae and this contributed to the high larval indices. However, improved sanitation conditions through the clean-up exercise during the restriction period in the community may have caused lower larval indices than what was observed in 2017 before the pandemic. Multiple insecticide resistance couple with three *kdr* mutations among the *Ae. aegypti* population in Cape Coast could affect the effectiveness of control measures, especially in emergency situations. The study supports sanitation improvement as a tool to control *Ae. aegypti* [27,47] and could complement insecticide-based tools in controlling this vector.

## Supporting information

**S1 Table. The genotype frequencies of V1016I, V410L and F1534C *kdr* mutations in *Aedes aegypti* population from Cape Coast.**
(XLSX)

## Acknowledgments

We are very grateful to Mr Francis Abortri who facilitated community entry and helped in data collection during the larval survey. JA also obtained CIH One Health Scholarship. We also thank the Department of Conservation Biology and Entomology, at the University of Cape Coast for their support.

## Author Contributions

**Conceptualization:** Andreas A. Kudom.

**Data curation:** Joana Ayettey, Andreas A. Kudom.

**Formal analysis:** Andreas A. Kudom.

**Investigation:** Joana Ayettey, Andreas A. Kudom.

**Methodology:** Joana Ayettey, Aikins Ablorde, Godwin K. Amlalo, Ben A. Mensah, Andreas A. Kudom.

**Project administration:** Andreas A. Kudom.

**Resources:** Andreas A. Kudom.

**Supervision:** Ben A. Mensah, Andreas A. Kudom.

**Validation:** Andreas A. Kudom.

**Writing – original draft:** Joana Ayettey, Ben A. Mensah, Andreas A. Kudom.

**Writing – review & editing:** Ben A. Mensah, Andreas A. Kudom.

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
