## [Decision Letter · Decision Letter 0]

29 Oct 2022

Dear Dr Kudom,

Thank you very much for submitting your manuscript "Multiple kdr mutations in Aedes aegypti, and the risk of an outbreak of Aedes-borne diseases during the Covid-19 restriction period in Cape Coast, Ghana" for consideration at PLOS Neglected Tropical Diseases. As with all papers reviewed by the journal, your manuscript was reviewed by members of the editorial board and by several independent reviewers. In light of the reviews (below this email), we would like to invite the resubmission of a significantly-revised version that takes into account the reviewers' comments. 

We cannot make any decision about publication until we have seen the revised manuscript and your response to the reviewers' comments. Your revised manuscript is also likely to be sent to reviewers for further evaluation.

Sincerely,

Mariangela Bonizzoni

Academic Editor

Esther Schnettler

Section Editor

Reviewer's Responses to Questions

**Key Review Criteria Required for Acceptance?**

**Methods**

-Are the objectives of the study clearly articulated with a clear testable hypothesis stated?

-Is the study design appropriate to address the stated objectives?

-Is the population clearly described and appropriate for the hypothesis being tested?

-Is the sample size sufficient to ensure adequate power to address the hypothesis being tested?

-Were correct statistical analysis used to support conclusions?

-Are there concerns about ethical or regulatory requirements being met?

Reviewer #1: Please refer to Summary and General Comments

Reviewer #2: The study tests the hypothesis that mosquito indexes were lower due to the Covid pandemic.

The study uses only mosquito indexes to address their questions; they do not include precipitation or epidemiological data.

Due to the Covid pandemic, the authors say collecting data from other localities was challenging.

 It would be helpful to add a session on the Methods describing the statistical analysis (what software, tests, etc.)

Reviewer #3: This study had a clear objective. However, the authors did not provide a lot of information to answer the questions above.

**Results**

-Does the analysis presented match the analysis plan?

-Are the results clearly and completely presented?

-Are the figures (Tables, Images) of sufficient quality for clarity?

Reviewer #1: Please refer to Summary and General Comments

Reviewer #2: Yes.

Reviewer #3: The results have been provided but there is need to make improvement.

**Conclusions**

-Are the conclusions supported by the data presented?

-Are the limitations of analysis clearly described?

-Do the authors discuss how these data can be helpful to advance our understanding of the topic under study?

-Is public health relevance addressed?

Reviewer #1: Please refer to Summary and General Comments

Reviewer #2: The mosquito indexes were lower, and they concluded that it was due to the pandemic. However, they mention the limitation of the study area.

Reviewer #3: This has been done but needs more work to make it clear.

**Editorial and Data Presentation Modifications?**

Reviewer #1: (No Response)

Reviewer #2: “Minor Revision”

Reviewer #3: There are minor editorial revisions to be made.

**Summary and General Comments**

Reviewer #1: The research paper entitled “Multiple kdr mutations in Aedes aegypti, and the risk of an outbreak of Aedes-borne diseases during the Covid-19 restriction period in Cape Coast, Ghana” aims to analyze any effects of the Covid-19 restriction period (mostly due to increased water demand and storage thus increased breeding site for Aedes species) in certain entomological parameters. Having said that the authors conclude that the opposite happened, vector populations decreased due to sanitary measures that were taken. Although the concept is interesting there are several limitations that require extra experimental work or at the very least to be discussed:

1. It is mentioned lLine 256) that the study assessed the risk of transmission of Aedes-borne diseases. This is not true, but should have been performed (infection status- DENV/CHIKV etc of mosquitoes) to be honest as this is most relevant. 

2. Metabolic resistance, a major mechanism of Aedes aegypti resistance, was not studies neither at the phenotypic nor at the molecular levels. For example, the OP/carbamate resistance can not be attributed to kdr mutations.

Additional comments

3. The clean-up exercises, as they are mentioned, should be described in more, technical details, and discussed more elaborately to show how they could have helped the situation.

4. Rows 41-43, conclusions: this is not supported by the study’s data and is dubious to say the least; insecticides remain the mainstay of vector control. 

5. Introduction: This section would benefit from information regarding insecticide resistance and its mechanisms, at least for target site that was studied.

6. At attempt should be made to explain the differences in insecticide resistance (phenotypic and molecular) at the different sites studied. 

7. Line 306: Ae. aegypti is mentioned, but in Table 1, only Aedes genus is presented. Where mosquitoes identified to species level?

Reviewer #2: The manuscript is well-written and easy to follow. The experimental design is sufficient due to the Covid pandemic. It would be great if the authors could add meteorological or epidemiological data from the two periods they compared the mosquito indexes.

Line 28 -> change "to adults were tested against four insecticides; " to "to adults were tested against four insecticides: "

Would it be possible to obtain precipitation data from 2017 and 2020? Your data show a lower outbreak risk. Could it be due to lower precipitation and not directly associated with Covid measures? Please add anything indicating other potential factors regarding the lower levels in 2020 compared to 2017.

Table 1 shows that you collected a lot of larvae from concrete tanks (1,088 larvae), representing almost half of your total larvae collected in HWSC. Did you consider that your insecticide resistance data might is biased toward this group of larvae since you used F1s? How big were these tanks? Could it be that the larvae you collected from there were all related to one single female that laid eggs there? One could easily remove water bottles or smaller plastic containers during clean-ups during the Covid measures, but the large water reservoir would be impossible.

Would it be possible to add epidemiological data from 2017 and 2022 for Dengue, for example? The study's conclusions rely on lower indexes, but it would be great if the number of Dengue cases also decreased. I guess that people were not moving around too much, reducing the transmission of Dengue.

Figure 4 shows the compound genotype frequencies of each mutation. How did you "phase" them? Since you have heterozygous genotypes, it becomes impossible to determine which mutation is on each parental chromosome. For example, three loci can have up to eight three possible haplotypes.

Reviewer #3: 1. Rewrite the title to make it better.

2. Your response “All relevant data are within the manuscript” to the question “Describe where the data may be found in

3. full sentences……” is not sufficient. Provide your databases as supplementary files or links to where to access it. 

4. This research study was conducted from July to September 2020 when COVID-19 infections were high and movement restrictions in place. How did you obtain permission to conduct this study and what precautions did you put in place to prevent contributing towards COVID-19 transmissions?

5. Page 12 Lines 130-131, clearly describe how the pre-adult mosquito stages were sampled from the selected houses. 

6. Page 12 Lines 134-135, provide a brief description of House index (HI), Container index (CI) and Breteau index (BI) like HI was expressed as the percentage of houses infested with Aedes larvae; CI as the percentage of containers infested; and BI as the number of positive containers per 100 inspected houses. 

7. Did the house owners provide consent to sample their houses?

8. Did this research study obtain ethical approval? If yes, indicate the approval number and the Institutional Review Board (IRB) or Ethical Review Committee (ERC) that approved it.

9. Data analysis section is mission. Indicate how you managed and analyzed the data you collected.

10. Indicate how the spatial distribution map in Figure 1 was made.

11. Read the PLOS NTD Instructions to Authors on Figures and Tables and comply appropriately. 

12. Omit sub-titles in the discussion section. 

13. Did you ask how long water was stored in each of the containers before it was fully emptied?

14. Page 22 Lines 258-259, cite the previous study that was done in 2017 and provide a summary of its findings. 

15. With clear understanding of the ecology and behavior of Aedes aegypti, clearly discuss how important the multiple kdr mutations are to it.

16. In your discussion, find a goof link among multiple kdr mutations, risk of Ae. aegypti arboviral outbreaks and COVID-19 restrictions. 

17. Discuss the limitations of this study.

18. Make your conclusion brief and clear.

PLOS authors have the option to publish the peer review history of their article (what does this mean?). If published, this will include your full peer review and any attached files.

Reviewer #1: No

Reviewer #2: No

Reviewer #3: Yes: Bryson Alberto Ndenga
---

## [Decision Letter · Decision Letter 1]

10 Mar 2023

Dear Dr Kudom,

Thank you very much for submitting your manuscript "Entomological surveillance on Aedes aegypti during covid 19 period in Cape Coast, Ghana: risk of arboviral outbreaks, multiple insecticide resistance and distribution of F1534C, V410L and V1016I kdr mutations" for consideration at PLOS Neglected Tropical Diseases. As with all papers reviewed by the journal, your manuscript was reviewed by members of the editorial board and by several independent reviewers. The reviewers appreciated the attention to an important topic. Based on the reviews, we are likely to accept this manuscript for publication, providing that you modify the manuscript according to the review recommendations. 

Sincerely,

Mariangela Bonizzoni

Academic Editor

Esther Schnettler

Section Editor

Reviewer's Responses to Questions

**Key Review Criteria Required for Acceptance?**

**Methods**

-Are the objectives of the study clearly articulated with a clear testable hypothesis stated?

-Is the study design appropriate to address the stated objectives?

-Is the population clearly described and appropriate for the hypothesis being tested?

-Is the sample size sufficient to ensure adequate power to address the hypothesis being tested?

-Were correct statistical analysis used to support conclusions?

-Are there concerns about ethical or regulatory requirements being met?

Reviewer #1: (No Response)

**Results**

-Does the analysis presented match the analysis plan?

-Are the results clearly and completely presented?

-Are the figures (Tables, Images) of sufficient quality for clarity?

Reviewer #1: (No Response)

**Conclusions**

-Are the conclusions supported by the data presented?

-Are the limitations of analysis clearly described?

-Do the authors discuss how these data can be helpful to advance our understanding of the topic under study?

-Is public health relevance addressed?

Reviewer #1: (No Response)

**Editorial and Data Presentation Modifications?**

Reviewer #1: (No Response)

**Summary and General Comments**

Reviewer #1: The authors have adequately addressed most comments raised in the first round. The only recommendation would be to discuss the lack of monitoring for the infectious status of mosquitoes as a limitation of the study.

PLOS authors have the option to publish the peer review history of their article (what does this mean?). If published, this will include your full peer review and any attached files.

Reviewer #1: No

Figure Files:

Data Requirements:

Reproducibility:

References

---

## [Editor Report · Decision Letter 2]

29 Mar 2023

Dear Dr Kudom,

Thank you very much for submitting your manuscript "Entomological surveillance on Aedes aegypti during covid 19 period in Cape Coast, Ghana: risk of arboviral outbreaks, multiple insecticide resistance and distribution of F1534C, V410L and V1016I kdr mutations" for consideration at PLOS Neglected Tropical Diseases. 

We noted that your most recent resubmission did not contain a Response to Reviewers, and the tracked-changes manuscript did not appear to show any changes since the prior round of review. We would therefore like to request that you please address the outstanding reviewer request from the last round of review, "to discuss the lack of monitoring for the infectious status of mosquitoes as a limitation of the study," and provide a Response to Reviewers and tracked-changes manuscript file demonstrating how you have addressed this comment. 

Sincerely,

Mariangela Bonizzoni

Academic Editor

Esther Schnettler

Section Editor

Figure Files:

Data Requirements:

Reproducibility:

References

---

## [Editor Report · Decision Letter 3]

22 May 2023

Dear Dr Kudom,

We are pleased to inform you that your manuscript 'Entomological surveillance on Aedes aegypti during covid 19 period in Cape Coast, Ghana: risk of arboviral outbreaks, multiple insecticide resistance and distribution of F1534C, V410L and V1016I kdr mutations' has been provisionally accepted for publication in PLOS Neglected Tropical Diseases.

Best regards,

Mariangela Bonizzoni

Academic Editor

Esther Schnettler

Section Editor

---

## [Editor Report · Acceptance letter]

26 May 2023

Dear Dr Kudom,

We are delighted to inform you that your manuscript, "Entomological surveillance on *Aedes aegypti* during covid 19 period in Cape Coast, Ghana: risk of arboviral outbreaks, multiple insecticide resistance and distribution of F1534C, V410L and V1016I kdr mutations," has been formally accepted for publication in PLOS Neglected Tropical Diseases.

Best regards,

Shaden Kamhawi

co-Editor-in-Chief

Paul Brindley

co-Editor-in-Chief
